# Prevalence and associated factors of MAFLD in adults with type 2 diabetes

**Yifei He[1]◉, Feng Xiao[2]◉, Bin Yi[2]\*, Jin Lu[1]\***

**1** Department of Endocrinology, Changhai Hospital, Naval Medical University, Shanghai, People's Republic of China, **2** Department of Liver Transplantation, Eastern Hepatobiliary Surgery Hospital, Naval Medical University, Shanghai, People's Republic of China

◉ These authors contributed equally to this work.
\* lujin-sh@139.com (JL); billyyi11@163.com (BY)

## Abstract

To estimate the prevalence and associated factors of hepatic steatosis and fibrosis in adults with type 2 diabetes (T2DM) in the United States.Data were retrieved from the 2017–March 2020 prepandemic cycle of the National Health and Nutritional Examination and Survey (NHANES). The study population included patients with T2DM. The controlled attenuation parameter (CAP) and liver stiffness measurement (LSM) were used to assess hepatic steatosis and fibrosis, respectively. A total of 1,290 T2DM patients were included, 85.2% (1044 patients) of whom presented with hepatic steatosis (CAP>248 dB/m). Among the 1044 T2DM patients with metabolically associated fatty liver disease (MAFLD), 29.5% developed hepatic fibrosis (LSM>8 kPa). Non-Hispanic black individuals (adjusted OR = 0.4008), BMI (adjusted OR = 1.1627), HbA1c (adjusted OR = 1.1450), TG (adjusted OR = 1.2347), HDL (adjusted OR = 0.4981), ALT (adjusted OR = 1.0227), AST (adjusted OR = 0.9396), and albumin (adjusted OR = 1.7030) were independently associated with steatosis. Age (adjusted OR = 1.0300), female sex (adjusted OR = 0.6655), BMI (adjusted OR = 1.1324), AST (adjusted OR = 1.0483), and GGT (adjusted OR = 1.0101) were independently associated with fibrosis. Heart failure was an independent factor associated with advanced fibrosis (adjusted OR = 1.9129) and cirrhosis (adjusted OR = 2.228). In the United States, hepatic steatosis is highly prevalent among T2DM patients, with 29.5% of these patients developing hepatic fibrosis. Some components of metabolic syndrome are related to hepatic steatosis and fibrosis. Moreover, heart failure is an independent factor associated with advanced fibrosis and cirrhosis.

## Introduction

With the increasing incidence of obesity and metabolic syndrome worldwide, nonalcoholic fatty liver disease (NAFLD) has become the leading cause of chronic liver disease in developed regions [1]. NAFLD is most closely related to the mortality of chronic liver disease and is even the major indication for liver transplantation [2]. In the past 20 years, numerous studies have clearly shown that NAFLD is a liver manifestation of a systemic metabolic disorder and is, in fact, a metabolic disease [3]. Consequently, the existing name "NAFLD" has been replaced by a

nhanes database (https://www.cdc.gov/nchs/nhanes/index.htm), it's free public database.

**Funding:** This work was funded by a grant from Changhai Hospital (234 Discipline Climbing Plan) Grant ID: 2019YXK021 to JL. The funders had no role in study design, data collection and analysis, decision to publish, or preparation of the manuscript.

**Competing interests:** The authors have declared that no competing interests exist.

new landmark term, "metabolically associated fatty liver disease" (MAFLD) [4, 5]. The significant difference between MAFLD and NAFLD is that the diagnosis of MAFLD does not require the exclusion of alcohol consumption [6]. This implies that individuals with metabolic disorders, regardless of heavy alcohol intake or hepatitis virus infection, are classified under MAFLD. Consequently, the prevalence of MAFLD may be greater than that of NAFLD, warranting recalibration. In accordance with the diagnostic criteria for MAFLD, we conducted an analysis of prevalence statistics and associated factors related to this condition.

To be diagnosed with MAFLD, patients must first meet one or more of the following criteria: obesity or overweight, type 2 diabetes mellitus, and metabolic disorders. These conditions can be identified through blood biochemistry tests, anthropometric measurements, or medical history inquiries. Second, there must be evidence of steatosis or fibrosis in the liver. Liver biopsy is considered the gold standard for diagnosis [7], but it is an invasive operation with obvious complications and shortcomings. As a result, the following non-invasive methods have received considerable attention [8, 9]: (1) hematological tests (serum fibrosis markers, laboratory tests); (2) methods to assess the physical properties of liver tissue (liver stiffness, degree of attenuation, viscosity); and (3) assessment of imaging of the liver and other abdominal organ learning methods.

Transient elastography (TE) is the most widely used noninvasive examination in the United States [10]. It is performed quickly and can be easily repeated. Furthermore, meta-analyses have shown that TE results have significant prognostic value [11], and the FDA has approved it as a test to evaluate liver fibrosis. TE incorporates two physical parameters: the controlled attenuation parameter (CAP) [12] and liver stiffness measurement (LSM) [13]. The CAP is a promising rapid and standardized instantaneous detection index of hepatic steatosis [14]. It ranges from 100–400 dB/m. A large meta-analysis revealed that the three values of 248 dB/m, 268 dB/m and 280 dB/m represent 5–10%, 33%, and 66% steatosis, respectively. The sensitivity and specificity of these values are 69%, 77%, and 88% and 82%, 81%, and 78%, respectively [15]. Therefore, in our study, MAFLD was also defined as a CAP≥248 dB/m, 268 dB/m, and 280 dB/m, which are considered indicative of moderate hepatic steatosis and severe hepatic steatosis. LSM reflects liver hardness. In a systematic evaluation comparing TE and biopsy for the detection of severe liver fibrosis, the overall sensitivity and specificity were 82% and 86%, respectively [16]. LSM is also included in the examination of the NHANES database during the 2017–March 2020 prepandemic cycle, and 14,400 out of 15,660 people underwent this examination. The guidelines [17] recommend the use of an LSM>8 kPa to diagnose liver fibrosis, with 9.6 kPa and 13 kPa as the cutoff values for medium-term and late fibrosis, respectively. Our study also adopted these cutoff values [18].

The National Health and Nutritional Examination and Survey(NHANES) [19] is a nationally representative public database that collects considerable medical and health information about Americans, including demographic, socioeconomic, dietary and health-related issues. TE is a new project in the NHANES database. At present, no article describes the use of TE to assess the prevalence of MAFLD in patients with T2DM. Our study fills this gap. We also identified independent factors associated with the development of MAFLD in these patients.

## Study design and methods

This was a cross-sectional study. Population data were collected from the 2017–March 2020 pre-pandemic cycle of the NHANES; these data are the latest data currently available in the database. The data for the NHANES 2019–2020 cycle were incomplete, as data collection was suspended due to the start of the coronavirus epidemic in 2019; therefore, these data were integrated into the 2017–2018 cycle to form the 2017–March 2020 prepandemic cycle. The

Research Ethics Review Board of the Centers for Disease Control and Prevention approved all surveys and medical examinations, and the respondents provided written informed consent.

A total of 15,560 individuals participated in the surveys and medical examinations, and we excluded participants who did not undergo TE (N = 5277) or MEC (N = 1260) assessment, leaving 9,023 people. Among these people, 7733 individuals were excluded because they did not meet the criteria for a diagnosis of T2DM (HbA1c $\leq$ 6.5%, fasting blood glucose $\leq$ 7 mmol/l, other types of diabetes, age $\leq$18Y, and absence of a history of T2DM). In total, 1290 participants with T2DM were included in the study. In addition, we selected 1044 patients with both T2DM and MAFLD to calculate the prevalence of fibrosis in those individuals (Fig 1).

The basic characteristics, complications and habits were collected via questionnaire. The basic characteristics included age, sex, and race. The complications included hypertension, stroke, coronary heart disease, renal insufficiency, heart failure, and diabetic retinopathy. Habits included alcohol consumption.

The laboratory biochemical tests included metabolism-related indicators and liver sclerosis indicators. Glycosylated hemoglobin, (HbA1c), total cholesterol (TC), triglyceride (TG), high-density lipoprotein (HDL), and uric acid (UA) levels were measured and used as metabolism-related indicators. Alanine aminotransferase (ALT), aspartate aminotransferase (AST), glutamyl transpeptidase (GGT), platelet (PLT), and albumin levels were measured and used as liver sclerosis indicators. All biochemical blood tests were performed with patients in a fasted condition.

The anthropometric measurements used in our research included weight and height. Body mass index (BMI) was calculated as weight in kilograms divided by height in meters squared and then rounded to one decimal place.

## Liver ultrasound transient elastography

The NHANES Mobile Examination Centre (MEC) has used the FibroScan 502 V2 Touch model for TE examination since 2017. The machine is equipped with medium (M) or ultra-large (XL) rods (probes). To obtain accurate data, the TE test must meet all of the following criteria: subjects fasted for more than 3 hours, 10 or more complete hardness (E) measurements, and a liver hardness interquartile range (IQRe)/median E < 30%.

## Statistical analysis

All the statistical data were obtained with Empowerstats software (www.empowerstats.com; X&Y Solutions, Inc., Boston MA). Due to population bias, prevalence statistics were performed with NHANES MEC weights as suggested by the National Center for Health Statistics and were calculated via the following formula. To describe the basic information of the population, categorical variables are expressed as percentages, and the chi-square test was used to compare the differences between different groups. Continuous variables are expressed as the mean ± standard deviation, and linear regression analysis was used to compare the differences between different groups. After adjusting for confounding factors, logistic regression was used to analyze the effects of different variables on hepatic steatosis and hepatic fibrosis. p <0.05 was considered to indicate a statistically significant difference.

$$w_i, MEC = w_i(base, MEC)f_i(NR, MEC)f_i(TR, MEC)f_i(C, MEC) = w_i(base, screener)$$
$$f_i(NR, screener)f_i(TR, screener)$$

$$f_i(NR, interview)f_i(TR, interview)f_i(C, interview)f_i(NR, MEC)f_i(TR, MEC)f_i(C, MEC)$$

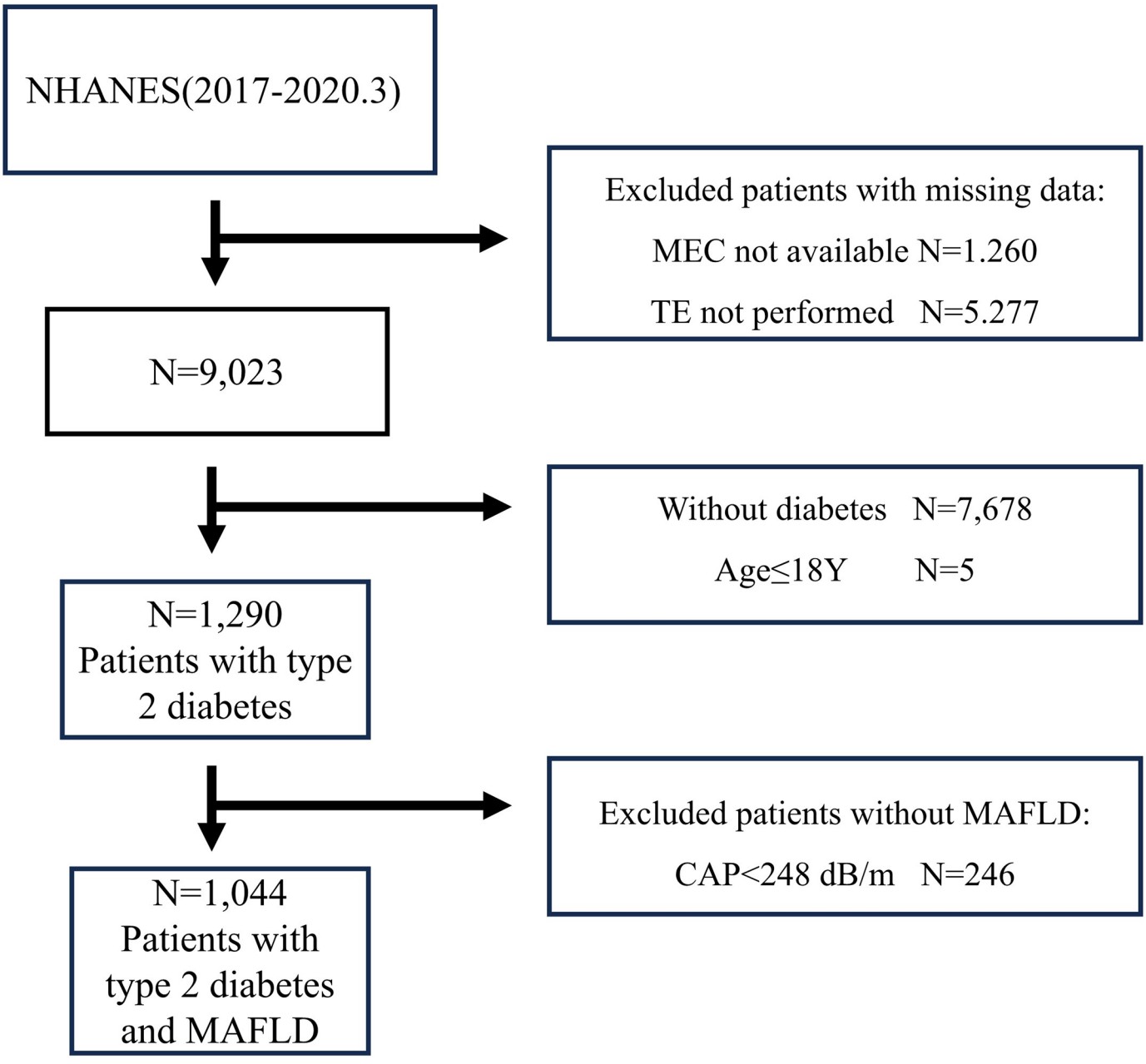

**Fig 1. Flow chart of the study.**

## Results

### Weighted prevalence of hepatic steatosis and fibrosis

In this study, a total of 1290 T2DM patients were screened from 15560 Americans. According to the CAP cut-off in Fig 2A, these patients were divided into four groups: nonadipose tissue (S0, CAP≤248 dB/m), mild steatosis (S1, 248 dB/m<CAP≤268 dB/m), moderate steatosis (S2, 268 dB/m<CAP≤280 dB/m) and severe steatosis (S3, CAP>280 dB/m). A total of

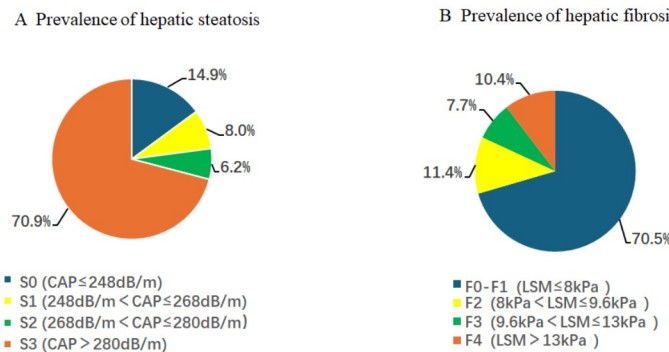

**Fig 2. Prevalence of hepatic steatosis and fibrosis.** A. Controlled attenuation parameter (CAP) data of patients. B. Liver stiffness measurement (LSM) data of patients.

1044 patients had hepatic steatosis (S1–S3), the weighted prevalence rate was 85.2%, and the weighted prevalence rates of S1-, S2- and S3-grade steatosis were 8%, 6.2%, and 70.9%, respectively. As shown in Fig 2B, 1044 T2DM patients with hepatic steatosis (S1–S3) were divided into four groups based on LSM values: mild fibrosis (F0–F1, LSM≤8 kPa), significant fibrosis (F2, 8 kPa<LSM≤9.6 kPa), advanced fibrosis (F3, 9.6 kPa<LSM≤13 kPa) and cirrhosis (F4, LSM>13 kPa). The overall weighted prevalence of fibrosis (F2-F4) was estimated to be 29.5%, and the weighted prevalence rates of grades F2 and F3 fibrosis were 11.4% and 7.7%, respectively. In total, 91 patients developed cirrhosis (F4), with a weighted prevalence rate of 10.4%.

### Characteristics of the participants stratified by CAP

All participants (1290 T2DM patients) were divided into four groups based on their CAP values, as shown in Table 1. Age, BMI, HbA1c, TG, HDL, ALT, AST, GGT, albumin, occurrence of CVD, and occurrence of HF were significantly different (all P < 0.05). In addition, the proportion of Mexican Americans and non-Hispanic whites increased with increasing steatosis, whereas the proportion of non-Hispanic blacks was the opposite.

### Characteristics of the participants stratified by LSM

All 1044 T2DM patients with MAFLD were divided into four groups based on their LSM values, as shown in Table 2. Among the different groups, BMI, HDL, UA, ALT, AST, GGT, PLT, occurrence of stroke, and occurrence of HF were significantly different (all P < 0.05).

### Factors associated with steatosis and fibrosis

As shown in Table 3, we used CAP>248 dB/m as the cutoff point for hepatic steatosis. Logical analysis revealed that BMI (adjusted OR = 1.1627, 95% CI: 1.1245, 1.2023), HbA1c (adjusted OR = 1.1450, 95% CI: 1.0267, 1.2770), TG (adjusted OR = 1.2347, 95% CI: 1.0183, 1.4971), and ALT (adjusted OR = 1.0227 CI: 1.0070, 1.0387) were positively associated with hepatic steatosis, while non-Hispanic black individuals (adjusted OR = 0.4008, 95% CI: 0.2273, 0.7067), HDL (adjusted OR = 0.4981, 95% CI: 0.3024, 0.8206), and AST (adjusted OR = 0.9396, 95% CI: 0.9139, 0.9660) were negatively associated with hepatic steatosis. As shown in Table 4, we used LSM>8 kPa as the cutoff point for hepatic fibrosis; age (adjusted OR = 1.0300, 95% CI: 1.0149, 1.0452), BMI (adjusted OR = 1.1324, 95% CI: 1.1037, 1.1619), AST (adjusted

**Table 1. Features of the study population according to CAP values.**

| | Total (n = 1290) | CAP (dB/m) | | | | P value |
|---|---|---|---|---|---|---|
| | | ≤248 (n = 246) | 248–268 (n = 125) | 268–280 (n = 81) | >280 (n = 838) | |
| Age(years) | 60.6± 12.9 | 64.2 ± 13.2 | 61.7 ± 11.5 | 62.7 ± 13.7 | 59.5 ± 12.8 | <0.0001 |
| Gender(%) | | | | | | 0.0817 |
| Male | 54.2 | 55.7 | 42.5 | 51.1 | 55.4 | |
| Female | 45.8 | 44.3 | 57.5 | 48.9 | 44.6 | |
| Race(%) | | | | | | 0.0002 |
| Mexican American | 9.9 | 6.1 | 6.8 | 10.5 | 11 | |
| Other Hispanic | 7.6 | 7.6 | 10.3 | 4.4 | 7.5 | |
| Non-Hipanic White | 56.5 | 49.7 | 51.2 | 59.5 | 58.2 | |
| Non-Hipanic Black | 13.4 | 24.3 | 19.5 | 14.9 | 10.2 | |
| Other Race | 12.7 | 12.3 | 12.2 | 10.6 | 13 | |
| BMI(kg/m2) | 33.0±7.1 | 27.5 ± 5.4 | 31.1 ± 6.4 | 30.3 ± 4.9 | 34.6 ± 7.0 | <0.0001 |
| HbA1c(%) | 7.3 ± 1.5 | 6.9 ± 1.3 | 7.1 ± 1.6 | 7.0 ± 1.3 | 7.5 ± 1.6 | <0.0001 |
| TC(mmol/l) | 4.6 ± 1.2 | 4.5 ± 1.2 | 4.4 ± 1.2 | 4.5 ± 1.0 | 4.6 ± 1.2 | 0.2231 |
| TG(mmol/l) | 1.7 ± 1.4 | 1.1 ± 0.5 | 1.6 ± 1.6 | 1.5 ± 0.7 | 1.9 ± 1.5 | <0.0001 |
| HDL(mmol/l) | 1.2 ± 0.3 | 1.4 ± 0.4 | 1.2 ± 0.4 | 1.2 ± 0.3 | 1.2 ± 0.3 | <0.0001 |
| UA(mmol/l) | 333.0 ± 93.6 | 325.2 ± 100.1 | 319.9 ± 86.6 | 332.2 ± 71.7 | 336.2 ±94.4 | 0.2315 |
| ALT(U/L) | 24.9 ± 17.7 | 17.9 ± 10.6 | 19.5 ± 14.3 | 21.9 ± 15.3 | 27.3 ± 18.8 | <0.0001 |
| AST(U/L) | 22.1 ± 13.0 | 19.2 ± 9.2 | 20.2 ± 10.6 | 20.2 ± 8.0 | 23.1 ± 14.1 | 0.0004 |
| GGT(U/L) | 38.3 ± 45.6 | 28.7 ± 34.6 | 34.1 ± 58.7 | 29.6 ± 17.7 | 41.6 ± 47.2 | 0.0008 |
| PLT (10$^9$/L) | 241.8 ± 68.7 | 238.42± 66.9 | 250.3 ± 71.6 | 231.7± 60.2 | 242.7 ±69.7 | 0.2819 |
| Albumin (g/dl) | 4.00± 0.34 | 4.0 ± 0.4 | 4.0± 0.4 | 4.13 ± 0.36 | 3.9 ± 0.33 | 0.0038 |
| Hypertension(%) | 67.9 | 68.2 | 63.3 | 62 | 68.9 | 0.7486 |
| CVD(%) | 12.2 | 12.8 | 3.8 | 6.8 | 13.5 | 0.0017 |
| HF(%) | 7.12 | 3.4 | 4.9 | 0.6 | 8.7 | 0.0237 |
| Stroke(%) | 9.26 | 10.6 | 10.5 | 8.6 | 8.9 | 0.5301 |
| CKD(%) | 7.4 | 9.5 | 7.5 | 6.0 | 7.1 | 0.6060 |
| DR(%) | 18.4 | 23.9 | 18.7 | 25.2 | 16.8 | 0.0772 |
| Alcohol(%) | 90.4 | 90.7 | 96.8 | 90.7 | 89.5 | 0.1676 |

BMI-body mass index;HbA1c-glycosylated hemoglobin;TC-total cholesterol;TG-triglyceride;HDL-high density lipoprotein;UA-uric acid;ALT-talanine aminotransferase;AST-aspartate aminotransferase; GGT-glutamyltranspeptidase;PLT-platelet;CVD-coronary;HF -heart failure;CKD-renal insufficiency; DR-diabetes retinopathy

OR = 1.0483, 95% CI: 1.0198, 1.0776), and GGT (adjusted OR = 1.0101, 95% CI: 1.0057, 1.0144) were positively associated with hepatic fibrosis, whereas female sex (adjusted OR = 0.6655, 95% CI: 0.4713, 0.9398) was negatively associated with hepatic fibrosis.

As shown in Table 5, LSM values of 9.6 kPa and 13 kPa were used as the thresholds for advanced fibrosis (F3) and cirrhosis (F4), respectively. Logistic regression analysis revealed that the occurrence of heart failure in patients with advanced fibrosis (F3) was 1.9129 times greater (adjusted OR = 1.9129, 95% CI: 1.0771, 3.3974) than those without advanced fibrosis. Furthermore, individuals with liver cirrhosis (F4) presented a 2.2289-fold increase in the occurrence of heart failure (adjusted OR = 2.2289, 95% CI: 1.0900, 4.5578). Additionally, other factors, such as BMI, ALT, AST, and GGT, were identified as independent predictors of both advanced fibrosis (F3) and cirrhosis (F4).

**Table 2. Features of the study population according to LSM values.**

| | Total (n = 1044) | LSM (kPa) | | | | P value |
|---|---|---|---|---|---|---|
| | | ≤8 (n = 106) | 8–9.6 (n = 94) | 9.6–13 (n = 81) | >13 (n = 763) | |
| Age(years) | 59.8 ± 12.7 | 60.0 ± 12.8 | 58.3 ± 11.4 | 59.6 ± 14.2 | 60.1 ± 11.5 | 0.6090 |
| Gender(%) | | | | | | 0.1829 |
| Male | 53.93 | 51.9 | 61.7 | 36.7 | 57 | |
| Female | 46.07 | 48.1 | 38.3 | 43.3 | 43 | |
| Race(%) | | | | | | 0.5030 |
| Mexican American | 10.73 | 10.6 | 10.4 | 12.9 | 10.3 | |
| Other Hispanic | 7.66 | 7.4 | 6.5 | 12.2 | 7.5 | |
| Non-Hipanic White | 57.18 | 56.2 | 61.6 | 53 | 62.7 | |
| Non-Hipanic Black | 11.60 | 13 | 9.4 | 11.6 | 4.6 | |
| Other Race | 12.82 | 12.8 | 12.6 | 10.3 | 15 | |
| BMI(kg/m2) | 40.0 ± 6.9 | 32.7 ± 6.3 | 35.9 ± 6.5 | 36.8 ± 6.8 | 39.1 ± 8.8 | <0.0001 |
| HbA1c(%) | 7.4 ± 1.6 | 7.4 ± 1.6 | 7.6 ± 1.6 | 7.5 ± 1.7 | 7.3 ± 1.3 | 0.4010 |
| TC(mmol/l) | 4.6 ± 1.2 | 4.6 ± 1.2 | 4.6 ± 1.2 | 4.3 ± 1.1 | 4.7 ± 1.2 | 0.1902 |
| TG(mmol/l) | 1.8 ± 1.5 | 1.8 ± 1.3 | 2.1 ± 2.2 | 2.0 ± 2.1 | 1.6 ± 1.0 | 0.2777 |
| HDL(mmol/l) | 1.2 ± 0.3 | 1.2 ± 0.3 | 1.1 ± 0.3 | 1.1 ± 0.3 | 1.1 ± 0.3 | 0.0118 |
| UA(mmol/l) | 334.3 ± 92.3 | 327.8 ± 93.7 | 341.4 ± 79.5 | 344.9 ±102.5 | 360.4 ±84.6 | 0.0031 |
| ALT(U/L) | 26.2 ± 18.3 | 23.9 ± 15.4 | 31.6 ± 17.7 | 30.1 ± 18.5 | 33.2 ± 29.7 | <0.0001 |
| AST(U/L) | 22.6± 13.5 | 20.5 ± 8.5 | 25.8 ± 11.8 | 25.4 ± 14.4 | 31.0 ± 28.8 | <0.0001 |
| GGT(U/L) | 40.0 ± 47.0 | 33.0 ± 32.5 | 45.9 ± 47.6 | 52.7 ± 54.4 | 71.7 ± 89.1 | <0.0001 |
| PLT (10$^9$/L) | 242.3 ± 69.1 | 246.0 ± 69.4 | 236.8 ± 68.9 | 246.2 ± 64.4 | 223.3 ± 68.9 | 0.0101 |
| Albumin (g/dl) | 4.0 ± 0.3 | 4.0 ± 0.3 | 4.0 ± 0.3 | 4.0 ± 0.3 | 4.0 ± 0.4 | 0.2930 |
| Hypertension(%) | 68.1 | 65.8 | 72.5 | 58.1 | 68.3 | 0.3794 |
| CVD(%) | 12.2 | 11.6 | 8.0 | 20.1 | 13.4 | 0.1545 |
| HF(%) | 7.9 | 6.5 | 5.1 | 5.6 | 21.8 | <0.0001 |
| Stroke(%) | 8.9 | 9.2 | 7.2 | 5.6 | 11.3 | 0.004 |
| CKD(%) | 7.1 | 6.70 | 4.32 | 13.3 | 8.04 | 0.2263 |
| DR(%) | 17.3 | 17.1 | 11.1 | 25.7 | 18.8 | 0.3618 |
| Alcohol(%) | 90.3 | 90.0 | 91.5 | 97.2 | 85.6 | 0.0761 |

BMI-body mass index;HbA1c-glycosylated hemoglobin;TC-total cholesterol;TG-triglyceride;HDL-high density lipoprotein;UA-uric acid;ALT-talanine aminotransferase;AST-aspartate aminotransferase; GGT-glutamyltranspeptidase;PLT-platelet;CVD-coronary;HF -heart failure;CKD-renal insufficiency; DR-diabetes retinopathy

## Discussion

In our research, all participants were diagnosed with T2DM, and MAFLD was diagnosed when hepatic steatosis was present. Using CAP>248 dB/m as the criterion for hepatic steatosis and LSM> 8 kPa as the criterion for significant fibrosis, we observed that the weighted prevalence of MAFLD in patients with T2DM was notably high at 85.2%, which is higher than the prevalence of NAFLD reported by Li Cho et al. [20]. They conducted a meta-analysis on different diagnostic tools and different ethnic groups to calculate the prevalence of NAFLD in T2DM patients and reported that the highest prevalence was diagnosed by biopsy, reaching 95%, followed by TE, which is similar to our method used in this study, reaching 75%. When stratified by ethnicity, the prevalence of NAFLD in T2DM patients in the Americas was 70%. They also reported a nonalcoholic steatohepatitis (NASH) prevalence of 31.55%, which is not significantly different from our reported 29%.

**Table 3. Analysis of the different variables involved in the occurrence of hepatic steatosis.**

| Variables | OR1 | Non-adjusted | P value | OR2 | Adjusted* | P value |
|---|---|---|---|---|---|---|
| Age(years) | 0.9732 | (0.9616, 0.9849) | <0.0001 | 0.9962 | (0.9815, 1.0111) | 0.2786 |
| Gender(%) | | | | | | |
| Male | 1 | | | 1 | | |
| Female | 1.0716 | (0.8107, 1.4164) | 0.6271 | 1.1582 | (0.8174, 1.6411) | 0.4087 |
| Race | | | | | | |
| Mexican American | 1 | | | | | |
| Other Hispanic | 0.7065 | (0.3781,1.3202) | 0.2760 | 0.7919 | (0.39431.5904) | 0.5119 |
| Non-HipanicWhite | 0.8249 | (0.4842,1.4054) | 0.4788 | 0.9373 | (0.5150,1.7060) | 0.8321 |
| Non-Hipanic Black | 0.3624 | (0.2198,0.5978) | <0.0001 | 0.4008 | (0.2273,0.7067) | 0.0015 |
| Other Race | 0.6029 | (0.3457,1.0513) | 0.0744 | 1.0576 | (0.5689,1.9658) | 0.8595 |
| BMI(kg/m2) | 1.1662 | (1.1320, 1.2014) | <0.0001 | 1.1627 | (1.1245, 1.2023) | <0.0001 |
| HbA1c(%) | 1.1729 | (1.0641, 1.2930) | 0.0013 | 1.1450 | (1.0267, 1.2770) | 0.0149 |
| TC(mmol/l) | 1.1312 | (1.5142, 6.9236) | 0.0527 | 1.0087 | (0.8547, 1.1904) | 0.9187 |
| TG(mmol/l) | 1.8299 | (1.5006, 2.2314) | <0.0001 | 1.2347 | (1.0183, 1.4971) | 0.0320 |
| HDL(mmol/l) | 0.2473 | (0.1669, 0.3665) | <0.0001 | 0.4981 | (0.3024, 0.8206) | 0.0062 |
| UA(mmol/l) | 1.0015 | (0.0213, 0.0913) | 0.0209 | 0.9983 | (0.9980, 1.0017) | 0.3009 |
| ALT(U/L) | 1.0373 | (1.0227, 1.0522) | <0.0001 | 1.0227 | (1.0070, 1.0387) | 0.0045 |
| AST(U/L) | 1.0101 | (0.9967, 1.0237) | 0.1399 | 0.9396 | (0.9139, 0.9660) | <0.0001 |
| GGT(U/L) | 1.0062 | (1.0014, 1.0110) | 0.0109 | 1.0007 | (0.9096, 1.0040) | 0.9965 |
| PLT(109/L) | 1.0025 | (1.0004, 1.0046) | 0.0209 | 1.0013 | (0.9988, 1.0038) | 0.3009 |
| Albumin (g/dl) | 1.2142 | (0.8054, 1.8305) | 0.3541 | 1.7030 | ((1.0335, 2.806) | 0.0366 |
| Hypertension | 0.8755 | (0.6476, 1.1837) | 0.3875 | 0.8964 | (0.6024, 1.2548) | 0.4548 |
| CVD | 0.9976 | (0.6245, 1.5936) | 0.9919 | 0.8491 | (0.4442, 1.6229) | 0.6923 |
| HF | 1.2220 | (0.6899, 2.1643) | 0.4918 | 1.3084 | (0.6712, 2.5506) | 0.4299 |
| Stroke | 0.5867 | (0.3804,0.9050) | 0.0158 | 0.7314 | (0.4366, 1.2252) | 0.2346 |
| CKD | 0.6248 | (0.4012,0.9731) | 0.0374 | 0.8897 | (0.5115, 1.5473) | 0.6788 |
| DR | 0.6245 | (0.4369,0.8928) | 0.0098 | 0.7089 | (0.4654,1.0798) | 0.1091 |
| Alcohol | 1.2499 | (0.8051, 1.9404) | 0.3203 | 1.5261 | (0.9174, 2.5386) | 0.1035 |

BMI-body mass index;HbA1c-glycosylated hemoglobin;TC-total cholesterol;TG-triglyceride;HDL-high density lipoprotein; UA-uric acid;ALT-talanine aminotransferase;AST-aspartate aminotransferase; GGT-glutamyltransptidase;PLT-platelet;CVD-coronary;HF -heart failure;CKD-renal insufficiency; DR-diabetes retinopathy.

*Models were adjusted for age, sex,race.BMI,HbA1c,TG,HDL,ALT, GGT

In the analysis of factors associated with MAFLD, we demonstrated the association of obesity, high blood glucose, high TG, and low HDL with MAFLD. Among these factors, BMI showed the strongest correlation with hepatic steatosis, and these results were consistent with those of previous studies [21, 22] and even showed a greater correlation in the study by Leite et al. [23] (OR: 7.1, 95% CI: 3.0–17.0). There was a slight difference in the incidence of hypertension. Hypertension is widely recognized as a risk factor for MAFLD in the general population [24]. However, several studies [21, 22] based on the NHANES database and our research did not find a correlation between blood pressure and MAFLD. This may be related to the population selected for the NHANES.

Among the metabolic factors related to liver fibrosis, obesity was the only factor that was significantly and independently correlated with liver fibrosis; this close relationship may be attributed to common pathophysiological insulin resistance mechanisms. Kwok et al. [25] also reported this association. Additionally, the authors suggested that the LSM was related to the

**Table 4. Analysis of the various variables involved in the occurrence of hepatic fibrosis.**

| Variables | OR1 | Non-adjusted | P value | OR2 | Adjusted* | P value |
|---|---|---|---|---|---|---|
| Age(years) | 0.9942 | (0.9835, 1.0050) | 0.2912 | 1.0300 | (1.0149, 1.0452) | <0.0001 |
| Gender(%) | | | | | | |
| Male | 1 | | | 1 | | |
| Female | 0.7499 | (0.5688, 0.9886) | 0.0412 | 0.6655 | (0.4713, 0.9398) | 0.0207 |
| Race | | | | | | |
| Mexican American | 1 | | | | | |
| Other Hispanic | 1.0701 | (0.6292, 1.8199) | 0.8026 | 1.0408 | (0.5725, 1.8922) | 0.8956 |
| Non-HipanicWhite | 1.1100 | (0.7221, 1.70635) | 0.6341 | 0.9158 | (0.5588, 1.5009) | 0.7272 |
| Non-Hipanic Black | 1.0705 | (0.5328, 1.3162) | 0.4417 | 0.6690 | (0.3952, 1.1326) | 0.1345 |
| Other Race | 0.6568 | (0.6643, 1.7251) | 0.7794 | 1.5609 | (0.9037, 2.6961) | 0.1102 |
| BMI(kg/m2) | 1.0944 | (1.0726, 1.1167) | 0.6341 | 1.1324 | (1.1037, 1.1619) | <0.0001 |
| HbA1c(%) | 1.0612 | (0.9801, 1.1491) | 0.4417 | 1.0537 | (0.9537, 1.1642) | 0.3040 |
| TC(mmol/l) | 0.9126 | (0.8096, 1.0288) | 0.1346 | 0.9293 | (0.7953, 1.0859) | 0.3559 |
| TG(mmol/l) | 1.1002 | (1.0192, 1.1876) | 0.0143 | 1.0189 | (0.9233, 1.1245) | 0.7092 |
| HDL(mmol/l) | 0.4738 | (0.2971, 0.7556) | 0.0017 | 0.6642 | (0.3648, 1.2093) | 0.1807 |
| UA(mmol/l) | 1.0026 | (1.0012, 1.0041) | 0.0003 | 1.0007 | (0.9989, 1.0024) | 0.4571 |
| ALT(U/L) | 1.0226 | (1.0148, 1.0305) | <0.0001 | 0.9900 | (0.9720, 1.0084) | 0.2841 |
| AST(U/L) | 1.0463 | (1.0323, 1.0605) | <0.0001 | 1.0483 | (1.0198, 1.0776) | 0.0008 |
| GGT(U/L) | 1.0129 | (1.0091, 1.0168) | <0.0001 | 1.0101 | (1.0057, 1.0144) | <0.0001 |
| PLT(109/L) | 0.9988 | (0.9968, 1.0008) | 0.2364 | 1.0000 | (0.9976, 1.0024) | 0.9995 |
| Albumin(g/dl) | 0.6792 | (0.4532, 1.0178) | 0.0608 | 0.9011 | (0.5381, 1.5089) | 0.6921 |
| Hypertension | 1.2604 | (0.9359, 1.6976) | 0.1275 | 1.1624 | (0.8120, 1.6641) | 0.4109 |
| CVD | 1.1383 | (0.7259, 1.7851) | 0.5724 | 0.9391 | (0.5556, 1.5874) | 0.8145 |
| HF | 1.6187 | (0.9955, 2.6319) | 0.4109 | 1.2960 | (0.7476, 2.2466) | 0.4109 |
| Stroke | 1.1824 | (0.6895, 1.8437) | 0.5383 | 1.4005 | (0.7665, 2.3953) | 0.2533 |
| CKD | 1.2468 | (0.7672, 2.0260) | 0.3732 | 1.2385 | (0.7086, 2.1646) | 0.4527 |
| DR | 0.9003 | (0.6024, 1.3455) | 0.6083 | 1.0718 | (0.6780, 1.6941) | 0.7667 |
| Alcohol | 1.1407 | (0.7095, 1.8341) | 0.5867 | 0.6951 | (0.4000, 1.2081) | 0.1972 |

BMI-body mass index;HbA1c-glycosylated hemoglobin;TC-total cholesterol;TG-triglyceride;HDL-high density lipoprotein; UA-uric acid;ALT-talanine aminotransferase;AST-aspartate aminotransferase; GGT-glutamyltranspeptidase;PLT-platelet;CVD-coronary;HF -heart failure;CKD-renal insufficiency; DR-diabetes retinopathy.

*Models were adjusted for age, sex,race.BMI,HbA1c,TG,HDL,ALT, GGT

duration of diabetes, ALT level, urinary albumin/creatinine ratio, and HDL-C level. However, Joseph [26] reported that advanced fibrosis (F3) and cirrhosis (F4) were not associated with obesity but were associated with diabetes duration and HbA1c. The variation in findings may be attributed to the limited data on liver fibrosis in T2DM patients and the fact that these studies were single-center studies. Additionally, it could also be influenced by differences in ethnicities or different states of diabetes, as observed in Joseph and Raymond's study involving an Asian population.

We also observed that women were less likely to develop MAFLD, while increasing age promoted the development of liver fibrosis. These results are consistent with the risk factors for NAFLD in the general population [27–29]. In addition, ALT, AST, and GGT are recognized as predictors of steatosis and fibrosis.

Limited research has been conducted on the association between liver fibrosis and heart failure in the diabetic population. The findings of this study confirmed an independent and

**Table 5. Analysis of the various variables involved in the occurrence of advanced fibrosis (F3) and cirrhosis (F4).**

| Variables | OR[1] | Adjusted* | P value | OR[2] | Adjusted* | P value |
|---|---|---|---|---|---|---|
| BMI | 1.1205 | (1.0911, 1.1508) | <0.0001 | 1.1218 | (1.0849, 1.1600) | <0.0001 |
| HDL | 0.5949 | (0.2969, 1.1918) | 0.1429 | 0.8077 | (0.3070, 2.1254) | 0.6653 |
| ALT | 1.0244 | (1.0150, 1.0339) | <0.0001 | 1.0238 | (1.0133, 1.0343) | <0.0001 |
| UA | 0.9999 | (0.9979, 1.0019) | 0.9154 | 0.9997 | (0.9969, 1.0024) | 0.8055 |
| AST | 1.0776 | (1.0401, 1.1164) | <0.0001 | 1.0537 | (1.0250, 1.0833) | 0.0002 |
| GGT | 1.0087 | (1.0050, 1.0123) | <0.0001 | 1.0109 | (1.0067, 1.0151) | <0.0001 |
| PLT | 0.9982 | (0.9954, 1.0011) | 0.2266 | 0.9951 | (0.9909, 0.9993) | 0.0217 |
| HF | 1.9129 | (1.0771, 3.3974) | 0.0268 | 2.2289 | (1.0900, 4.5578) | 0.0280 |
| Stroke | 1.3019 | (0.6790, 2.4961) | 0.4269 | 1.6984 | (0.7407, 3.8944) | 0.2109 |

BMI-body mass index;HDL-high density lipoprotein;UA-uric acid;ALT-talanine aminotransferase;AST-aspartate aminotransferase; GGT-glutamyltranspeptidase; HF-heart failure;OR1, Logistic regression analysis for binary classification with LSM value of 6.9 kPa; OR2, Logistic regression analysis for binary classification with LSM value of 13kPa.

*Models were adjusted for age, sex,race.BMI,HbA1c,TG,HDL,ALT, GGT

positive relationship between advanced fibrosis (F3), cirrhosis (F4) and heart failure in this patient cohort, even after controlling for potential confounding factors. A study by Yanjian Wang [30] used the noninvasive biomarker Fibrosis NASH Index (FNI) to assess liver fibrosis and reported a significantly increased risk of HF with pEF (OR 1.59, 95% CI 1.22–2.08) in patients with type 2 diabetes who also had advanced liver fibrosis.

The aforementioned studies on the relationship among liver fibrosis, cirrhosis and heart failure were all cross-sectional studies, which were unable to determine the causal relationship among these conditions. We believe that they share common causative factors and are part of a pathophysiological continuum [31]. The pathogenesis of diabetes, such as the release of inflammatory factors, insulin resistance, and oxidative stress, can affect the development and severity of fatty liver disease and may directly promote the occurrence of heart disease and heart failure. In a hyperglycemic state, the release of inflammatory factors such as IL-6, TNF-α, IL-8, and IL-1β can activate hepatic stellate cells and Kupffer cells, causing liver fibrosis [32], and can also stimulate cardiac fibroblasts, exacerbating myocardial fibrosis [33]. Insulin resistance is the core pathogenesis of MAFLD [34] and simultaneously leads to disorders of glucose and lipid metabolism, promoting atherosclerosis. ROS are the main products of oxidative stress. Excessive ROS production leads to the activation of hepatic stellate cells, causing liver fibrosis [35]. It can also promote the transformation of macrophages into foam cells, which is a key step in the progression of atherosclerotic lesions [36].

In addition to the common factors mentioned above, these two diseases can also mutually promote each other [37]. The mechanism by which heart failure leads to liver fibrosis is related to liver congestion and hypoxia-reperfusion injury. During heart failure, the pressure in the superior and inferior vena cava increases, resulting in the obstruction of hepatic vein outflow and causing congestive liver disease [38]. This leads to an increase in hepatic sinus pressure, hepatocyte necrosis, and central lobular fibrosis, ultimately leading to cirrhosis. Another possible situation is that hypoxia causes continuous low perfusion, resulting in ischemic necrosis of hepatocytes [39]. The increased risk of heart failure due to liver fibrosis is associated with the exacerbation of insulin resistance in diabetic patients with NAFLD and the increase in pro-inflammatory factors, pro-fibrotic and vasoactive mediators; these factors can promote the development of cardiac and arrhythmic complications [40]. Additionally, liver fibrosis is

related to cardiac diastolic dysfunction [41], and the mechanism may be related to the increase in epicardial fat thickness [42].

Although these are the latest data extracted from the NHANES and MAFLD has been redefined according to the latest diagnostic criteria, there are still some limitations in our research. First, the cutoff points of CAP and LSM in elastography have not yet been unified [43], leading to variations in morbidity calculations. In addition, only five subjects were infected with hepatitis B or C, so we did not analyze them. In addition, we found no correlation between alcohol consumption and MAFLD when the quantity of alcohol consumed was used as a continuous variable. Finally, prospective studies are needed to clarify the causal relationships, such as those between BMI and heart failure in relation to liver fibrosis.

In conclusion, the data from the NHANES database represent the characteristics of the American population and show that patients with T2DM have a greater prevalence of MAFLD than NAFLD. In these populations, poor blood sugar control, high TG, low HDL, and overweight or obesity are associated factors for MAFLD. Furthermore, our study revealed that only obesity is an independent factor associated with liver fibrosis. Additionally, we observed a significant association between heart failure and advanced fibrosis (F3) and liver cirrhosis (F4). These findings serve as reminders that clinical physicians should pay timely attention to the impact of heart failure on liver fibrosis in MAFLD patients.

## Author Contributions

**Conceptualization:** Yifei He, Feng Xiao.

**Data curation:** Yifei He, Feng Xiao.

**Formal analysis:** Yifei He.

**Funding acquisition:** Jin Lu.

**Investigation:** Yifei He, Feng Xiao, Jin Lu.

**Methodology:** Yifei He, Feng Xiao, Bin Yi.

**Project administration:** Feng Xiao, Bin Yi, Jin Lu.

**Resources:** Bin Yi, Jin Lu.

**Software:** Feng Xiao, Jin Lu.

**Supervision:** Bin Yi, Jin Lu.

**Validation:** Feng Xiao, Jin Lu.

**Visualization:** Feng Xiao, Bin Yi.

**Writing – original draft:** Yifei He.

**Writing – review & editing:** Bin Yi, Jin Lu.

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
