## [Decision Letter · Decision Letter 0]

17 May 2024

PGPH-D-24-00927

Prevalence and Factors Associated with MAFLD in Adults with Type 2 Diabetes (T2DM)

Dear Bin Yi,

Thank you for submitting your manuscript to PLOS Global Public Health. After careful consideration, we feel that it has merit but does not fully meet PLOS Global Public Health’s publication criteria as it currently stands. Therefore, we invite you to submit a revised version of the manuscript that addresses the points raised during the review process.

We look forward to receiving your revised manuscript.

Kind regards,

Collins Otieno Asweto, PhD

Academic Editor

Journal Requirements:

2. Please provide separate figure files in .tif or .eps format only and remove any figures embedded in your manuscript file. Please also ensure all files are under our size limit of 10MB.

3. We do not publish any copyright or trademark symbols that usually accompany proprietary names, eg  ©, ®, ™  (e.g. next to drug or reagent names). Please remove all instances of trademark/copyright symbols throughout the text, including ® on page 6.

Additional Editor Comments (if provided):

Reviewers' comments:

Reviewer's Responses to Questions

**Comments to the Author**

1. Does this manuscript meet PLOS Global Public Health’s publication criteria? Is the manuscript technically sound, and do the data support the conclusions? The manuscript must describe methodologically and ethically rigorous research with conclusions that are appropriately drawn based on the data presented.

Reviewer #1: Yes

Reviewer #2: No

Reviewer #3: No

2. Has the statistical analysis been performed appropriately and rigorously?

Reviewer #1: Yes

Reviewer #2: No

Reviewer #3: Yes

3. Have the authors made all data underlying the findings in their manuscript fully available (please refer to the Data Availability Statement at the start of the manuscript PDF file)?

Reviewer #1: Yes

Reviewer #2: No

Reviewer #3: No

4. Is the manuscript presented in an intelligible fashion and written in standard English?

Reviewer #1: Yes

Reviewer #2: No

Reviewer #3: No

5. Review Comments to the Author

Reviewer #1: General comments:

• This cross-sectional study has very interesting findings for clinical practice and future studies. I congratulate the authors for their work on the predictors of steatosis and fibrosis amongst adult T2DM patients with MAFLD. The statistical analysis was great, the findings and discussions were well presented.

• There are grammatical errors that need correction to ensure readability and clarity. The entire manuscript requires serious proofreading.

o Page 5: under introduction, rephrase reference [1] by removing the ‘’.’’ Between syndrome and NAFLD and replace with a comma ‘’,’’. Also write NAFLD in full (Non-Alcoholic Fatty Liver Disease).

o Page 6: reference 6 should be rephrase e.g: To diagnose MAFLD, patients must first meet one or more of the following conditions: obesity or overweight, diabetes mellitus, metabolic disorders, which can be determined through blood biochemistry tests, anthropometric measurements, or medical history inquiries. Secondly, there must be evidence of steatosis or fibrosis in the liver. Liver biopsy remains the gold standard for diagnosis…….

o Page 7: include ‘’respectively’’ after 78% in reference 14 in relation to sensitivity and specificity.

o Check: Alcoal(%)

o Page 19, line 3: correct ‘’stuties’’ to ‘’studies’’ in relation to references 23 and 24 and also correct repetition after reference 27-29.

o Page 20, line 13 needs correction of the typo

o There is no need to have ‘’ Funding None. Declaration of Competing Interest The authors declare that they have no known competing financial interests or personal relationships that could have appeared to influence the work reported in this paper.’’ After the conclusion since it is to be reflected in the matrix on page 1.

• The lingering issue is the relationship between advanced fibrosis/cirrhosis and heart failure (causality or a consequence) and the pathogenesis/pathophysiology involved.

Reviewer #2: The manuscript is technically deficient as the authors do not know what they are doing based on what is written in the abstract (logistic regression) and statistical analysis (linear regression). The weighted analysis is not described well. It also has many grammatical errors that could have been checked before submitting.

Reviewer #3: The study to look in to: Prevalence and factors associated with MAFLD in adults with T2DM- the flow and presentation including justification, what is known and what the authors found is not clear. As it is a cross-sectional study, it is difficult to validate the findings presented as association between MAFLD and other factors. The association presented with increased prevalence of HF and Liver Fibrosis in adults with T2DM is not presented clearly to justify and validate the results.

It was not easy to read as the grammar and language including spellings need revision.

6. PLOS authors have the option to publish the peer review history of their article (what does this mean?). If published, this will include your full peer review and any attached files.

**Do you want your identity to be public for this peer review?** For information about this choice, including consent withdrawal, please see our Privacy Policy.

Reviewer #1: No

Reviewer #2: **Yes: **Shital Bhandary

Reviewer #3: **Yes: **Somasundari Gopalakrishnan

---

## [Decision Letter · Decision Letter 1]

16 Sep 2024

PGPH-D-24-00927R1

Prevalence and Factors Associated with MAFLD in Adults with Type 2 Diabetes (T2DM)

Dear Bin

Thank you for submitting your manuscript to PLOS Global Public Health. After careful consideration, we feel that it has merit but does not fully meet PLOS Global Public Health’s publication criteria as it currently stands. Therefore, we invite you to submit a revised version of the manuscript that addresses the points raised during the review process.

We look forward to receiving your revised manuscript.

Kind regards,

Collins Otieno Asweto, PhD

Academic Editor

Journal Requirements:

Reviewers' comments:

Reviewer's Responses to Questions

**Comments to the Author**

1. If the authors have adequately addressed your comments raised in a previous round of review and you feel that this manuscript is now acceptable for publication, you may indicate that here to bypass the “Comments to the Author” section, enter your conflict of interest statement in the “Confidential to Editor” section, and submit your "Accept" recommendation.

Reviewer #4: All comments have been addressed

Reviewer #5: All comments have been addressed

2. Does this manuscript meet PLOS Global Public Health’s publication criteria? Is the manuscript technically sound, and do the data support the conclusions? The manuscript must describe methodologically and ethically rigorous research with conclusions that are appropriately drawn based on the data presented.

Reviewer #4: Yes

Reviewer #5: No

3. Has the statistical analysis been performed appropriately and rigorously?

Reviewer #4: Yes

Reviewer #5: Yes

4. Have the authors made all data underlying the findings in their manuscript fully available (please refer to the Data Availability Statement at the start of the manuscript PDF file)?

Reviewer #4: Yes

Reviewer #5: No

5. Is the manuscript presented in an intelligible fashion and written in standard English?

Reviewer #4: Yes

Reviewer #5: No

6. Review Comments to the Author

Reviewer #4: The findings for the study are well represented and all issues raised by previous reviewers have been adequately addressed. However, the paper has some typos and minimal texts errors that need to be reviewed for clarity a simple proof reading should be able to resolve the issues. I have no further issues with the study.

Reviewer #5: This is an interesting study and the authors are well appreciated for the idea and the statistical analysis. However, the manuscript still needs thorough revision.

1. The introduction section lacks clear idea of what the research problem and gap are .

2. The objective of the study as mentioned, is not limited to Type 2 diabetes thus, the objective should include the association of MAFLD with variables other than Type 2 DM as observed in the statistical analysis and discussion section (eg. Hypertention, cardiac failure etc.)

3. The authors should clearly mention the nature of NHANES survey (cross-sectional or cohort) in the methodology section, as the authors try to establish a causal relationship of MAFLD with diabetes and other factors. If the survey was cross-sectional avoid making causal inferences. Also, justify including Liver function test parameters like ALT and AST as test variables. Raised ALT and AST are not the cause of MAFLD but an indicator of liver disease. There are several other variables whose inclusion into the study as a cause of MAFLD are not justified in the manuscript eg. GGT, Platelets count, HF, stroke etc.

4. The authors should justify that patients who were not classified as having Type 2 diabetes based on their HbA1c and fasting blood glucose levels were not actually cases of controlled diabetes.

5. Please justify why individuals with type 1 DM were excluded from the study.

6. The discussion needs to be re-written to be in line with the objective of the study.

7. Please avoid using the terms prevalence, or incidence in the results and discussion. The authors can rather recommend a randomized trial for causal inference based on the findings of the study.

7. There are numerous errors related grammatical and sentence structure. The whole manuscript needs english language expert editing.

7. PLOS authors have the option to publish the peer review history of their article (what does this mean?). If published, this will include your full peer review and any attached files.

**Do you want your identity to be public for this peer review?** For information about this choice, including consent withdrawal, please see our Privacy Policy.

Reviewer #4: No

Reviewer #5: No

---

## [Decision Letter · Decision Letter 2]

5 Nov 2024

Prevalence and Factors Associated with MAFLD in Adults with Type 2 Diabetes (T2DM)

PGPH-D-24-00927R2

Dear Bin yi,

We are pleased to inform you that your manuscript 'Prevalence and Factors Associated with MAFLD in Adults with Type 2 Diabetes (T2DM)' has been provisionally accepted for publication in PLOS Global Public Health.

Best regards,

Collins Otieno Asweto, PhD

Academic Editor

Reviewer's Responses to Questions

**Comments to the Author**

1. If the authors have adequately addressed your comments raised in a previous round of review and you feel that this manuscript is now acceptable for publication, you may indicate that here to bypass the “Comments to the Author” section, enter your conflict of interest statement in the “Confidential to Editor” section, and submit your "Accept" recommendation.

Reviewer #1: All comments have been addressed

Reviewer #5: All comments have been addressed

2. Does this manuscript meet PLOS Global Public Health’s publication criteria? Is the manuscript technically sound, and do the data support the conclusions? The manuscript must describe methodologically and ethically rigorous research with conclusions that are appropriately drawn based on the data presented.

Reviewer #1: Yes

Reviewer #5: Yes

3. Has the statistical analysis been performed appropriately and rigorously?

Reviewer #1: Yes

Reviewer #5: Yes

4. Have the authors made all data underlying the findings in their manuscript fully available (please refer to the Data Availability Statement at the start of the manuscript PDF file)?

Reviewer #1: Yes

Reviewer #5: Yes

5. Is the manuscript presented in an intelligible fashion and written in standard English?

Reviewer #1: Yes

Reviewer #5: Yes

6. Review Comments to the Author

Reviewer #1: Congratulations to the team for responding to all the reviewers' comments and improving the write up for focus and clarity.

Reviewer #5: All comments by the reviewer have been addressed. Considerable improvement in the language has been observed however, there are still some typological and grammatical errors throughout the manuscript.

7. PLOS authors have the option to publish the peer review history of their article (what does this mean?). If published, this will include your full peer review and any attached files.

**Do you want your identity to be public for this peer review?** For information about this choice, including consent withdrawal, please see our Privacy Policy.

Reviewer #1: No

Reviewer #5: No
